# Prompt-Based Exemplar Super-Compression and Regeneration for Class-Incremental Learning

## Abstract

Replay-based methods in class-incremental learning (CIL) have attained remarkable success. Despite their effectiveness, the inherent memory restriction results in saving a limited number of exemplars with poor diversity. In this paper, we introduce PESCR, a novel approach that substantially increases the quantity and enhances the diversity of exemplars based on a pre-trained general-purpose diffusion model, without fine-tuning it on target datasets or storing it in the memory buffer. Images are compressed into visual and textual prompts, which are saved instead of the original images, decreasing memory consumption by a factor of 24. In subsequent phases, diverse exemplars are regenerated by the diffusion model. We further propose partial compression and diffusion-based data augmentation to minimize the domain gap between generated exemplars and real images. Comprehensive experiments demonstrate that PESCR significantly improves CIL performance across multiple benchmarks, e.g., 3.2% above the previous state-of-the-art on ImageNet-100.

## 1 Introduction

Ideally, AI systems should be adaptive to ever-changing environments—where the data are continuously observed. The AI models should be capable of learning concepts from new data while maintaining the ability to recognize the old ones. In practice, AI systems often have constrained memory budgets, because of which most of the historical data must be abandoned. However, deep AI models suffer from catastrophic forgetting when being updated by abundant new data and limited historical data, as previous knowledge can be overridden by the new information (McCloskey & Cohen, 1989; Ratcliff, 1990). To study how to overcome catastrophic forgetting, the class-incremental learning (CIL) protocol (Rebuffi et al., 2017) is established. CIL assumes training samples from various classes are introduced to the model in phases, with previous data mostly discarded from memory.

CIL has enjoyed significant advancements (Kirkpatrick et al., 2017; Zenke et al., 2017; Aljundi et al., 2018; Chaudhry et al., 2018; Lee et al., 2017; Yoon et al., 2017; Yan et al., 2021; Wang et al., 2022a; Zhou et al., 2022; Pham et al., 2021), among which replay-based methods (Rebuffi et al., 2017; Hou et al., 2019; Liu et al., 2021a;b; Yan et al., 2021; Liu et al., 2020) stand out in terms of performance by employing a memory buffer to store a limited number of representative samples (i.e., exemplars) from former classes. During subsequent learning phases, these exemplars are revisited to help retain previously learned knowledge.

Although replay-based methods demonstrate notable effectiveness, they are still restricted by two main drawbacks. Firstly, since the exemplar set is much smaller compared to the new training data, the model is biased towards the new classes. Secondly, the poor exemplar diversity leads to overfitting on old classes. These two issues are essentially incurred by the lack of quantity and diversity of exemplars respectively. Therefore, by tackling these two problems, all replay-based CIL methods can potentially be enhanced.

We consider these questions in CIL: is it efficient to save old-class information as RGB images? Can we compress the images into something more compact so that more information can be stored with the same memory consumption? In this paper, we propose a novel approach named **P**rompt-based **E**xemplar **S**uper-**C**ompression and **R**egeneration (PESCR). Instead of directly storing the previous images, we compress them into visual and textual prompts, e.g., edge maps and textual descriptions, and save these prompts in

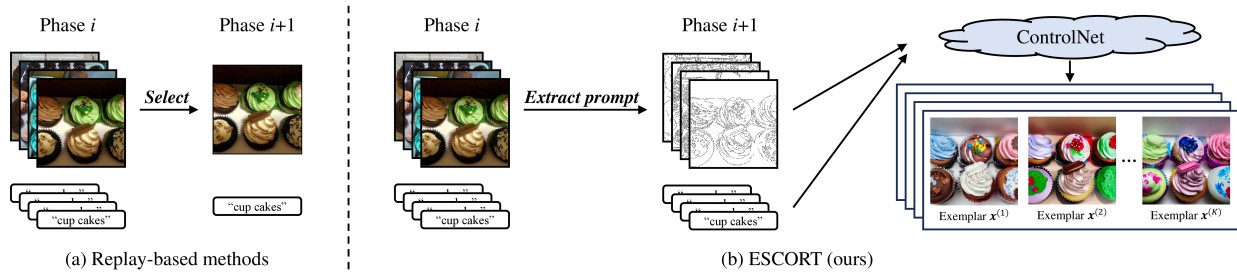

(a) Replay-based methods           (b) ESCORT (ours)

Figure 1: Comparison between traditional replay-based CIL methods with our approach. (a) **Traditional replay-based CIL methods** can only select and save a small number of exemplars due to the memory restriction, leading to two severe issues: firstly, the relatively small size of the exemplar set compared to the new training dataset gives rise to a pronounced imbalance between old and new classes; secondly, the limited diversity of the exemplar set compared to the original training set incurs an overfitting problem. (b) **PESCR** compresses the old images into visual and textual prompts, e.g., edge maps and class tags, and saves these prompts to the memory. In subsequent phases, diverse high-resolution exemplars are regenerated from these prompts via an off-the-shelf pre-trained diffusion model, e.g., ControlNet. PESCR dramatically improves the quantity and diversity of exemplars without violating the memory constraint.

the memory. In subsequent phases, diverse high-resolution exemplars are regenerated from the prompts by leveraging an off-the-shelf pre-trained diffusion model, e.g., ControlNet (Zhang et al., 2023).

Compared to traditional direct replay methods, PESCR enjoys increased **quantity** and enhanced **diversity** of the exemplar set, as illustrated in Figure 1. Firstly, the exemplar quantity is boosted by compression: since the memory consumption of a 1-bit edge map is merely $\frac{1}{24}$ that of its 8-bit, 3-channel RGB image counterpart, 24 times more exemplars can be saved within the same memory budget. We call this process super-compression as this compression ratio is far beyond that of existing image compression algorithms, without even compromising the image quality. Secondly, instead of relying on the images only from the dataset itself, we apply diffusion-based image generation to produce unseen samples with great diversity. This can be achieved simply by changing the random seed of the diffusion model.

However, utilizing generated images for CIL model training leads to a potentially huge domain gap between old exemplars and new data. We propose two techniques to mitigate this problem, i.e., partial compression and diffusion-based data augmentation, enabling the CIL model to properly benefit from the synthetic exemplars without the need to fine-tune the diffusion model on the target dataset. Since the same pre-trained diffusion model can be directly downloaded from the public cloud at any time when necessary, we do not need to store the fine-tuned generator using our own memory.

Extensive experiments show that our PESCR achieves top performance on both fine-grained and coarse-grained classification datasets: Caltech-256 (Griffin et al., 2022), Food-101 (Bossard et al., 2014), Places-100 (Zhou et al., 2016b), and ImageNet-100 (Deng et al., 2009). We fully investigate the effect of PESCR under different CIL settings and demonstrate that our approach achieves tremendous improvements compared to the state-of-the-art (SOTA) CIL method, e.g., substantially increasing the average learning from half accuracy on 11-phase ImageNet-100 by 3.2%. Our contributions can be summarized as follows.

- We challenge the traditional manner of saving old class exemplars as RGB images in CIL and propose a memory-efficient data storage approach based on prompts, significantly increasing exemplar quantity with the same memory cost.

- We employ a general-purpose ControlNet, without fine-tuning it on our target datasets, to regenerate diverse high-resolution images from prompts during incremental stages.

- We devise two techniques, i.e., partial compression and diffusion-based data augmentation, to alleviate the domain gap between generated exemplars and real images.

- We conduct extensive experiments on four classification datasets, two CIL protocols, seven CIL methods, and three budget settings to evaluate the performance of our approach.

## 2 Related Work

**Class-incremental learning.** The goal of class-incremental learning is to develop machine learning models that can effectively adapt to and learn from data presented in a series of training stages. This is closely associated with topics known as continual learning (De Lange et al., 2019a; Lopez-Paz & Ranzato, 2017) and lifelong learning (Chen & Liu, 2018; Aljundi et al., 2017). Recent incremental learning approaches are either task-based, i.e., all-class data come but are from a different task for each new phase (Shin et al., 2017; Zhao et al., 2020; Li & Hoiem, 2017), or class-based, i.e., each phase has the data of a new set of classes coming from the identical dataset (Yoon et al., 2017; Yan et al., 2021; Wang et al., 2022a; Zhou et al., 2022; Pham et al., 2021). The latter is typically called class-incremental learning (CIL). CIL methods can be grouped into three main categories: replay-based, regularization-based, and parameter-based (De Lange et al., 2019b; Prabhu et al., 2020). **Replay-based** methods employ a memory buffer to store information from previous classes for rehearsal in later stages. Direct replay (Isele & Cosgun, 2018; Aljundi et al., 2019; Iscen et al., 2020; Bang et al., 2021; Liu et al., 2021b) saves representative images from the dataset, while generative replay (Shin et al., 2017; He et al., 2018; Hu et al., 2018; Zhu et al., 2021; Petit et al., 2023; Liu et al., 2021c) saves generators trained by previous images. These strategies are centered around selecting key samples, training generators, and effectively enhancing the classification model by utilizing a dataset that combines both exemplars and new data. **Regularization-based** methods (Kirkpatrick et al., 2017; Zenke et al., 2017; Aljundi et al., 2018; Chaudhry et al., 2018; Lee et al., 2017) incorporate regularization terms into the loss function to reinforce previously acquired knowledge while assimilating new data. **Parameter-based** methods (Yoon et al., 2017; Yan et al., 2021; Wang et al., 2022a; Zhou et al., 2022; Pham et al., 2021) allocate distinct model parameters to each incremental phase, aiming to avoid model forgetting that arises from overwriting parameters

**Exemplar compression.** Attempts have been made to compress the exemplars and reduce their memory consumption, so that exemplar quantity can be increased with the same memory buffer. MRDC (Wang et al., 2022b) employs JPEG (Wallace, 1992) to compress exemplars and studies the trade-off between quality and quantity of the compressed images. CIM (Luo et al., 2023) identifies foreground regions of exemplars by CAM (Zhou et al., 2016a) and downsamples the background to compress images. These approaches have two weaknesses. 1) The compression is lossy and the exemplar image quality degrades. 2) The compression ratio is data-dependent, as JPEG compression falls short with irregular patterns and non-smooth gradients, and background downsampling is inefficient with dominating foreground regions. Our PESCR, in contrast, guarantees high-resolution exemplars and a constant compression ratio of 24, independent of the image dataset we choose.

**Diffusion models.** Diffusion models function by progressively deteriorating the data, introducing Gaussian noise in incremental steps, and then learning to reconstruct the data by reversing the noise introduction process (Ho et al., 2020; Singer et al., 2022; Villegas et al., 2022). They have demonstrated notable success in producing high-resolution and photorealistic images from a range of conditional inputs, e.g., natural language descriptions, segmentation maps, and keypoints (Ho et al., 2020; 2022; Zhang et al., 2023). Recently, text-to-image diffusion models such as Stable Diffusion (Rombach et al., 2022) have been employed to enhance training data as well. SDDR (Jodelet et al., 2023) leverages a pre-trained Stable Diffusion model to generate extra exemplars from class tags for CIL, but image generation based only on class tags can hardly recover detailed information of images in that class.

**Diffusion models with spatial control.** The recent development (Zhang et al., 2023) of adding spatial conditioning controls to pre-trained text-to-image diffusion models, as seen in the work referred to as ControlNet, has garnered significant interest. ControlNet, a variation of the diffusion model, achieves this by integrating a pre-trained large-scale diffusion model, like Stable Diffusion (Rombach et al., 2022), with zero convolutions in the network. This allows ControlNet to create high-quality images from the input text and a visual prompt, which can be edges, depth, segmentation, or human pose representations. This extraordinary ability sheds light on a new potential data format for storing old-class exemplars. Since visual prompts such

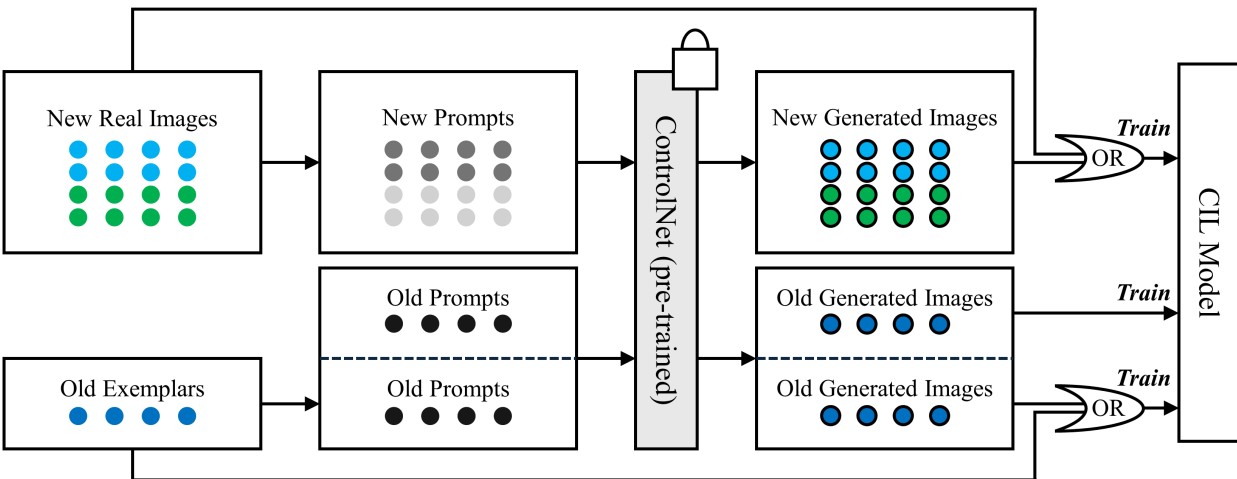

Figure 2: The CIL model training process with PESCR in the $i$-th phase ($i \geq 2$). A pre-trained ControlNet is downloaded in advance and remains frozen for image generation. Initially, we have three subsets of data because of partial compression: new real images of this phase, old prompts (i.e., edge maps and class tags), and old exemplars. Firstly, we transform the new real images and old exemplars into edge maps by Canny edge detection. Then, we use ControlNet to generate images from all the prompts we have. Finally, we train the CIL model with real and generated images. An image, if having both generated and real versions, will appear only once in each epoch, with a probability $p$ of being the generated version and $1 - p$ of being the real version.

as edge maps cost much less space than the original RGB images, more prompts can be saved with the same memory budget, and more high-resolution images with the same details can be regenerated by ControlNet in later stages for CIL model training.

## 3 Methodology

As illustrated in Figure 2, our approach generates diverse exemplars from former prompts to overcome the forgetting problem in CIL. Specifically, we describe the problem setup of replay-based CIL in Section 3.1. Then, we explain how to compress images to prompts in Section 3.2. Next, we show how to regenerate the exemplars from prompts and train the CIL model with them in Section 3.3. We introduce two techniques to reduce the domain gap in Section 3.4. The overall algorithm is provided in Section 3.5 and Algorithm 1.

### 3.1 Problem Setup

Reply-based CIL has multiple phases during which the number of classes gradually increases to the maximum (Douillard et al., 2020; Hou et al., 2019; Liu et al., 2020). In the 1-st phase, we observe data $\mathcal{D}_1$, using them to learn an initial model $\Theta_1$. After training, we can only store a small subset of $\mathcal{D}_1$ (i.e., exemplars denoted as $\mathcal{E}_1$) in memory used as replay samples in later phases. In the $i$-th phase ($i \geq 2$), we get new class data $\mathcal{D}_i$ and load exemplars $\mathcal{E}_{1:i-1} = \mathcal{E}_1 \cup \cdots \cup \mathcal{E}_{i-1}$ from the memory. Then, we initialize $\Theta_i$ with $\Theta_{i-1}$, and train it using $\mathcal{E}_{1:i-1} \cup \mathcal{D}_i$. We evaluate the model $\Theta_i$ on a test set $\mathcal{Q}_{1:i}$ with all classes observed so far. Eventually, we select exemplars $\mathcal{E}_{1:i}$ from $\mathcal{E}_{1:i-1} \cup \mathcal{D}_i$ and save them in the memory. For the growing memory buffer setup with a budget of $b$ units per class (where one unit corresponds to the memory cost of one image), we need to ensure that the final number of exemplars stored at this stage $|\mathcal{E}_{1:i}| \leq ib$. For the fixed memory buffer setup with a budget of $B$ units in total, we need to have $|\mathcal{E}_{1:i}| \leq B$ in all phases.

### 3.2 Prompt-Based Super-Compression

The performance of replay-based methods is severely limited by the quantity and diversity of exemplars. To tackle these two issues, we compress old images into visual and textual prompts and use them to regenerate the exemplars by utilizing an off-the-shelf diffusion model. As the prompts require much less memory compared to RGB images, we are able to save a large number of prompts within the same memory budget, so that far more exemplars can be regenerated in subsequent phases.

**Prompt extraction.** At the end of the $i$-th phase, we first randomly select a training instance $(\boldsymbol{x}, y)$ from the dataset $\mathcal{D}_i$, where $\boldsymbol{x}$ denotes the image and $y$ denotes the classification label.

Next, we extract the visual prompts. The visual prompts should preserve as many sufficient details as possible to help the CIL model retain the old-class knowledge. At the same time, they should be small enough to reduce the memory cost. Therefore, we choose Canny edge maps as the visual prompts. We use the classical Canny edge detector (Canny, 1986) to obtain the edge map $\boldsymbol{e}$ as follows:

$$\boldsymbol{e} = \text{CannyEdge}(\boldsymbol{x}). \tag{1}$$

Then, we directly save the class label $t$ as the textual prompt. For example, if the class label is "cupcakes", then $t =$ 'cupcakes'.

We repeat the above process to add other visual and textual prompts to the memory until the memory budget is exhausted. After that, we obtain the final prompt set $\mathcal{P}_i$ for the $i$-th phase, i.e., $\mathcal{P}_i = \{(\boldsymbol{e}_j, t_j)\}_{j=1}^{R_i}$, where $R_i$ denotes the maximum number of prompts at the $i$-th phase to fit in the memory.

**Prompt memory consumption.** Storing edge maps and class tags in the memory buffer takes far less space than storing the original images. For an 8-bit RGB image, compressing it to a 1-bit edge map of equal size achieves a compression ratio of $8 \times 3 = 24$. This means that each memory unit, which can originally store 1 exemplar image, can now store 24 edge maps. The class labels, which usually contain one or two words, consume negligible memory.

### 3.3 Exemplar Regeneration and CIL Model Training

To regenerate exemplars from past prompts, we leverage a general-purpose pre-trained ControlNet (Zhang et al., 2023), which can produce high-resolution images based on visual and textual prompts. We directly apply the off-the-shelf ControlNet without fine-tuning it on target datasets. Therefore, we do not need to allocate any memory space to save the ControlNet model, as we are able to re-download the ControlNet model from the cloud at the beginning of each phase. This setting has its strengths and weaknesses. By avoiding saving the generator in our buffer, we take full advantage of the memory to store old data. However, we have to sacrifice the generator's capability to fit our dataset. Consequently, the generated images might greatly differ from our images in various properties such as brightness, contrast, noise, etc., leading to a potentially huge domain gap. We propose two solutions to tackle this issue in Section 3.4.

**Exemplar regeneration.** In the $i$-th phase, we first take out a pair of visual and textual prompts $(\boldsymbol{e}, t)$ from the memory $\mathcal{P}_{1:i-1} = P_1 \cup \cdots \cup \mathcal{P}_{i-1}$ of $R_{1:i-1} = \sum_{m=1}^{i-1} R_m$ prompts in total. Then, we resize the edge map $\boldsymbol{e}$ with nearest neighbor interpolation to meet the input size requirement of ControlNet (i.e., both height and width must be a multiple of 64). After that, we forward the prompts $(\boldsymbol{e}, t)$ to ControlNet and generate $K$ new exemplars with $K$ different random seeds:

$$\hat{\boldsymbol{x}}_k = \text{ControlNet}(\boldsymbol{e}, t, s_k), \quad k = 1, \cdots, K, \tag{2}$$

where $k$ and $s_k$ are the index and its corresponding random seed, respectively. The generated image $\hat{\boldsymbol{x}}_k$ is then resized to the original size of $\boldsymbol{x}$ for consistency of image resolution. We repeat this operation until we finish processing the entire prompt set $\mathcal{P}_{1:i-1}$. Finally, we obtain the regenerated exemplar set $\hat{\mathcal{E}}_{1:i-1}$, which contains $R_{1:i-1}K$ exemplars.

**CIL model training.** Then, we combine the regenerated exemplars $\hat{\mathcal{E}}_{1:i-1}$ with the new training data $\mathcal{D}_i$ to train the CIL model:

$$\Theta_i \leftarrow \Theta_i - \gamma \nabla_{\Theta_i} \mathcal{L}(\Theta_i; \hat{\mathcal{E}}_{1:i-1} \cup \mathcal{D}_i), \tag{3}$$

where $\gamma$ denotes the learning rate, and $\mathcal{L}$ is the CIL training loss. To avoid bias towards a large number of similar regenerated images, in each epoch, we randomly sample only one image from the $K$ synthetic exemplars originating from the same base image for training.

### 3.4 Exemplar-Image Domain Gap Reduction

With PESCR, we can significantly increase the quantity and diversity of exemplars without breaking the memory constraint. However, the domain gap between the synthetic exemplars and real images is still a concern. Therefore, directly using the regenerated exemplars for CIL model training leads to suboptimal performance in practice.

A common approach to solve this problem is to fine-tune the generative model on the target dataset. However, if the generator is updated, we have to store it at the end of each phase, taking considerable memory of our buffer. To overcome this issue, we introduce the following two techniques to reduce the domain gap without fine-tuning ControlNet on our datasets.

**Partial compression.** It is indeed tempting to utilize all memory budget to save edge maps, as we would obtain 24 times the original number of exemplars, not to mention the $K$ random generations from each edge map. However, excessive exemplars incur a greater domain gap. Therefore, we consider only spending a part of our memory on saving edge maps, while leaving the rest to save RGB image exemplars as usual.

Specifically, for each phase $i$, we assume the memory budget at this phase is $B_i$ units in total, i.e., we are allowed to save at most $B_i$ images as exemplars normally. We set $\alpha$ as the compressed proportion of the dataset whose value can be adjusted depending on the CIL setting. So, only $\alpha B_i$ memory units will be allocated for edge maps, saving $24\alpha B_i$ edge maps; while the remaining $(1-\alpha)B_i$ units will be allocated for images, saving $(1-\alpha)B_i$ original images.

We firstly use herding (Rebuffi et al., 2017) to sort the training images in $\mathcal{D}_i$ by their representativeness. Then we save the $(1-\alpha)B_i$ most representative ones directly, while the next $24\alpha B_i$ images are stored as edge maps in the buffer. This preserves the most representative information of old classes in their original form. The remaining less representative images in $\mathcal{D}_i$ are discarded.

**Diffusion-based data augmentation.** Another technique we apply to attenuate the domain gap is diffusion-based data augmentation. During CIL model training, every real image $\boldsymbol{x}$ has a certain probability of being replaced by one of its $K$ generated copies.

Before training starts at the $i$-th phase, for each instance $(\boldsymbol{x}, y)$ with a real image $\boldsymbol{x}$, we extract the edge map $\boldsymbol{e}$ from $\boldsymbol{x}$ and obtain $K$ generated copies $\{\hat{\boldsymbol{x}}_k\}_{k=1}^K$ by Equation 2. In each training epoch, $\boldsymbol{x}$ has a probability $p$ of being replaced by any one of $\{\hat{\boldsymbol{x}}_k\}_{k=1}^K$ using uniform sampling. This augmentation operation enables the model to learn from generated features and mitigates the domain gap between real and synthetic images.

**Quantity–quality trade-off.** By adjusting the compressed proportion $\alpha$ and augmentation probability $p$, we can control the trade-off between the quantity and quality of generated information that is learned by the CIL model. Larger $\alpha$ and $p$ let the model learn from more generated exemplars more frequently, but the widened domain gap might cause performance degradation. Smaller $\alpha$ and $p$ alleviate the domain gap and improve the information quality, but the exemplar quantity and learning frequency are compromised. The optimal choice of $\alpha$ and $p$ depends on the CIL setting.

**Overall CIL training loss.** The CIL training process of the $i$-th phase (Equation 3) is adjusted to:

$$\Theta_i \leftarrow \Theta_i - \gamma \nabla_{\Theta_i} \mathcal{L}(\Theta_i; (\hat{\mathcal{E}}_{1:i-1} \vee \mathcal{E}_{1:i-1}) \cup (\hat{\mathcal{D}}_i \vee \mathcal{D}_i)), \tag{4}$$

where $\hat{\mathcal{E}}_{1:i-1}$ and $\mathcal{E}_{1:i-1}$ represent the regenerated subset and the real-image subset of previous exemplars, respectively. $\hat{\mathcal{D}}_i$ and $\mathcal{D}_i$ are the augmented version and the original version of the new dataset at phase $i$, respectively. $\vee$ denotes the logic OR operation.

---

**Algorithm 1:** CIL with PESCR (Phase $i$ with growing budget setup)

---

**Input:** New class data $\mathcal{D}_i$; old real-image exemplars $\mathcal{E}_{1:i-1}$; old prompts $\mathcal{P}_{1:i-1}$; CIL model $\Theta_{i-1}$;
random seeds $\{s_k\}_{k=1}^{K}$; compressed proportion $\alpha$, augmentation probability $p$, phase budget $B$

**Output:** New real-image exemplars $\mathcal{E}_i$; new prompts $\mathcal{P}_i$; CIL model $\Theta_i$

**1** Download the pre-trained ControlNet;

**2** # Get new generated images $\hat{\mathcal{D}}_i$ from new real images $\mathcal{D}_i$

**3** Set $\hat{\mathcal{D}}_i = \varnothing$;

**4 for** $(\boldsymbol{x}, y) \in \mathcal{D}_i$ **do**

**5** $\quad$ Get the prompts $(\boldsymbol{e}, t)$ from $(\boldsymbol{x}, y)$ by Equation 1;

**6** $\quad$ **for** $k = 1, \cdots, K$ **do**

**7** $\quad\quad$ Generate $\hat{\boldsymbol{x}}_k$ from $(\boldsymbol{e}, t, s_k)$ by Equation 2 and add $(\hat{\boldsymbol{x}}_k, y)$ to $\hat{\mathcal{D}}_i$;

**8** $\quad$ **end**

**9 end**

**10** # Get old generated images $\hat{\mathcal{E}}_{1:i-1}$ from old real images $\mathcal{E}_{1:i-1}$ and old prompts $\mathcal{P}_{1:i-1}$

**11** Set $\hat{\mathcal{E}}_{1:i-1} = \varnothing$;

**12 for** $(\boldsymbol{x}, y) \in \mathcal{E}_{1:i-1}$ **do**

**13** $\quad$ Get the prompts $(\boldsymbol{e}, t)$ from $(\boldsymbol{x}, y)$ by Equation 1;

**14** $\quad$ **for** $k = 1, \cdots, K$ **do**

**15** $\quad\quad$ Generate $\hat{\boldsymbol{x}}_k$ from $(\boldsymbol{e}, t, s_k)$ by Equation 2 and add $(\hat{\boldsymbol{x}}_k, y)$ to $\hat{\mathcal{E}}_{1:i-1}$;

**16** $\quad$ **end**

**17 end**

**18 for** $(\boldsymbol{e}, t) \in \mathcal{P}_{1:i-1}$ **do**

**19** $\quad$ **for** $k = 1, \cdots, K$ **do**

**20** $\quad\quad$ Generate $\hat{\boldsymbol{x}}_k$ from $(\boldsymbol{e}, t, s_k)$ by Equation 2 and add $(\hat{\boldsymbol{x}}_k, y)$ to $\hat{\mathcal{E}}_{1:i-1}$;

**21** $\quad$ **end**

**22 end**

**23** # Model training

**24** Initialize $\Theta_i$ with $\Theta_{i-1}$;

**25 for** iterations **do**

**26** $\quad$ Sample a training data point from $\hat{\mathcal{D}}_i \cup \hat{\mathcal{E}}_{1:i-1}$ with probability $p$ or from $\mathcal{D}_i \cup \mathcal{E}_{1:i-1}$ with probability $1 - p$;

**27** $\quad$ Update the model $\Theta_i$ by Equation 4;

**28 end**

**29** # Data storing

**30** Select the $(1 - \alpha)B$ most representative samples $(\boldsymbol{x}, y)$ from $\mathcal{D}_i$ to form $\mathcal{E}_i$;

**31** Select the next $24\alpha B$ representative $(\boldsymbol{x}, y)$ from $\mathcal{D}_i$ and add their prompts $(\boldsymbol{e}, t)$ to $\mathcal{P}_i$;

**32** Store $\mathcal{E}_i$ and $\mathcal{P}_i$ in the memory buffer.

---

### 3.5 Algorithm

In Algorithm 1, we summarize the overall procedures of the proposed PESCR in the $i$-th incremental learning phase ($i \geq 2$) with a growing budget setting of $B$ units per phase. Line 1 corresponds to the ControlNet preparation step. Lines 2-17 encapsulate the diffusion-based data augmentation for the training set. Lines 18-22 explicate the exemplar regeneration process. Lines 23-28 elucidate the CIL training procedures. (For minibatch updates, we just repeat this operation in each iteration to sample a minibatch.) Lines 29-32 explain the exemplar updating approach with partial compression.

Table 1: Average $N$-phase LFS and $(N + 1)$-phase LFH accuracies (%) of different methods, with $b = 5$ memory units/class for Caltech-256 and $b = 20$ for other datasets.

| CIL Method | Learning from Scratch (LFS) | | | | Learning from Half (LFH) | | | | | | | |
| | Caltech-256 | | ImageNet-100 | | Caltech-256 | | Food-101 | | Places-100 | | ImageNet-100 | |
| | $N$=5 | 10 | 5 | 10 | 5 | 10 | 5 | 10 | 5 | 10 | 5 | 10 |
|---|---|---|---|---|---|---|---|---|---|---|---|---|
| iCaRL (Rebuffi et al., 2017) | 57.7 | 48.9 | 66.2 | 58.7 | 53.4 | 49.1 | 58.6 | 50.0 | 42.2 | 37.6 | 59.2 | 50.3 |
| WA (Zhao et al., 2020) | 66.2 | 54.6 | 76.2 | 69.7 | 60.1 | 46.8 | 74.2 | 64.6 | 62.0 | 57.3 | 73.6 | 66.2 |
| MEMO (Zhou et al., 2022) | 65.7 | 61.4 | 78.5 | 74.0 | 62.6 | 60.1 | 71.1 | 50.1 | 53.4 | 48.8 | 72.5 | 70.6 |
| PODNet (Douillard et al., 2020) | 67.0 | 60.9 | 76.4 | 68.7 | 68.4 | 66.6 | 79.5 | 77.0 | 68.3 | 66.3 | 79.4 | 77.2 |
| FOSTER (Wang et al., 2022a) | 41.3 | 36.4 | 81.1 | 78.7 | 62.4 | 60.9 | 81.2 | 78.9 | 69.4 | 68.5 | 81.6 | 78.6 |
| CIM (Luo et al., 2023) | 65.5 | 66.3 | 82.0 | 78.0 | 64.1 | 65.5 | 79.5 | 77.1 | 71.1 | 70.5 | 80.5 | 79.5 |
| DER (Yan et al., 2021) | 68.1 | 64.8 | 81.8 | 78.5 | 68.4 | 66.8 | 82.0 | 80.4 | 70.4 | 69.5 | 81.8 | 80.2 |
| PESCR (ours) | **72.8** | **69.8** | **83.4** | **81.3** | **72.1** | **71.3** | **83.9** | **83.2** | **72.3** | **71.8** | **84.2** | **83.4** |

Table 2: Average 6-phase LFH accuracies (%) of PESCR on ImageNet-100, with $b = 20$ memory units/class. Diffusion-based augmentation is applied with probability $p$. The numbers $R + S$ in the first row indicate $R$ real and $S$ synthetic exemplars/class are saved in the buffer.

| $p$ | 20+0 | 19+24 | 18+48 | 17+72 | 16+96 | 15+120 | 14+144 | 13+168 | 12+192 |
|---|---|---|---|---|---|---|---|---|---|
| 0.0 | 81.8 | 82.4 | 82.4 | 82.5 | 82.8 | 82.6 | 81.6 | 81.4 | 80.8 |
| 0.1 | 82.4 | 82.9 | 83.4 | 83.4 | 83.5 | 83.4 | 83.8 | 83.4 | 83.5 |
| 0.2 | 82.8 | 83.0 | 83.3 | 83.6 | 83.9 | **84.2** | 83.7 | 83.4 | 83.9 |
| 0.3 | 82.5 | 83.4 | 83.6 | 84.1 | 83.8 | 84.0 | **84.2** | 84.0 | 84.0 |
| 0.4 | 82.9 | 83.1 | 83.5 | 84.0 | 83.7 | 83.9 | 83.8 | 84.0 | 83.6 |
| 0.5 | 82.4 | 82.2 | 83.1 | 82.8 | 83.5 | 83.5 | 83.4 | 83.7 | 83.4 |

## 4 Experiments

### 4.1 Experiment Settings

**Datasets.** We conduct CIL experiments and evaluate PESCR on four image classification datasets: Caltech-256, Food-101, Places-100, and ImageNet-100. **Caltech-256** (Griffin et al., 2022) is an object recognition dataset with 30,607 images from 257 classes (256 object classes and a clutter class), each class having 80 to 827 images. We remove the clutter class and keep at most 150 images in each class by random selection to avoid extreme class imbalance. The remaining images of each class are randomly split into training (80%) and test (20%) sets. **Food-101** (Bossard et al., 2014) contains 101,000 food images of 101 classes, each class with 750 training and 250 test images. **Places-100** is a subset of Places-365-Standard (Zhou et al., 2016b), a large-scale dataset including 365 scene categories with 3,068 to 5,000 training and 100 validation images per class. We construct the subset by randomly choosing 100 classes with seed 0. Then 3,000 training images are randomly chosen from each category for class balance. We use their validation set as the test set. **ImageNet-100** is a subset of ImageNet-1000 (Deng et al., 2009) randomly sampled with seed 1993, following (Hou et al., 2019), and each class has about 1,300 training and 50 test images.

**CIL protocols.** We adopt two protocols in our experiments: learning from half (LFH) and learning from scratch (LFS). LFH assumes the model is trained on half of the classes in the first phase and on the remaining classes evenly in the following $N$ phases. LFS assumes the model learns from an equal number of classes in each of the $N$ phases. We set $N$ to be 5 or 10 in our experiments. The model is evaluated on all the classes observed so far at each phase, and the final average classification accuracy is reported.

**Memory budget.** Although MEMO (Zhou et al., 2022) recently proposes a new buffer setting (with model memory taken into account), based on which MEMO attains SOTA performance by storing lighter models, we still follow the common setup, assigning a fixed number of $b$ memory units per class for all methods. The growing budget setting is more challenging than the fixed budget setting, where the memory in earlier phases is much more abundant. (As Caltech-256 has fewer images per class, we set $b = 5$ on Caltech-256 and $b = 20$ on other datasets by default, unless otherwise specified.)

Table 3: Average 11-phase LFH accuracies (%) on ImageNet-100 with $b = 20$ memory units/class, with and without PESCR plugged in. The accuracy improvements by applying PESCR are listed in the last row.

| | iCaRL | WA | MEMO | PODNet | FOSTER | DER |
|---|---|---|---|---|---|---|
| Baseline | 50.3 | 66.2 | 70.6 | 77.2 | 78.6 | 80.2 |
| PESCR | 66.4 | 71.7 | 77.6 | 80.6 | 79.4 | 83.4 |
| Improvements | +16.1 | +5.5 | +7.0 | +3.4 | +0.8 | +3.2 |

Table 4: Average 6-phase LFH accuracies (%) of PESCR on ImageNet-100 with $b = 20$ memory units/class and $K$ generated copies per image. Diffusion augmentation is applied with probability $p = 0.4$, and $R = 16$ real and $S = 96$ synthetic exemplars/class are saved in the buffer after partial compression.

| $K$ | 0 | 1 | 5 | 10 | 15 | 20 | 25 |
|---|---|---|---|---|---|---|---|
| Accuracy | 81.8 | 83.1 | 83.7 | 83.4 | **83.8** | 83.7 | 83.4 |

**Textual prompt extraction.** The textual prompt of each class is directly derived from its class label with minimal processing. For Caltech-256, we remove the prefix and suffix and replace the hyphen with a space. e.g., "063.electric-guitar-101" is changed to "electric guitar". For Food-101, we replace the underscore with a space. e.g., "apple_pie" is modified as "apple pie". For Places-100, which adopts a bi-level categorization scheme for some classes, such as "general_store/indoor", "general_store/outdoor", and "train_station/platform", we transform them to "indoor general store", "outdoor general store", and "train station platform" to ensure semantic meaningfulness. For ImageNet-100, the original class labels are used directly as textual prompts.

**Training setup.** All CIL algorithms are evaluated on ResNet-18 (He et al., 2016), which is trained by 200 epochs in the first phase and 170 epochs in subsequent phases with SGD. Data augmentations include random resized cropping, horizontal flip, color jitter, and AutoAugment (Cubuk et al., 2019), following (Wang et al., 2022a). We adopt the hyperparameters in PyCIL (Zhou et al., 2023) to implement all the CIL methods. The previous SOTA exemplar compression approach CIM (Luo et al., 2023) is also implemented by plugging into DER (Yan et al., 2021) and FOSTER (Wang et al., 2022a), and we report the better result for each experiment. Unless otherwise mentioned, we incorporate PESCR into DER (Yan et al., 2021), which generally has the best performance across various settings. We choose $\alpha$ and $p$ based on grid search and find that $\alpha \in [0.05, 0.3]$ and $p \in [0.2, 0.4]$ work the best in general. We generate $K = 5$ synthetic copies per image for diffusion-based augmentation, but we do not train the model with more epochs.

### 4.2 Results and Discussion

**Comparison with previous approaches.** We test the CIL performance of different methods with LFS and LFH protocols, illustrating the results in Table 1. PESCR significantly enhances the baseline method DER by a large margin: in 10-phase LFS setting, PESCR improves accuracy by 5.0% and 2.8% on Caltech-256 and ImageNet-100, respectively; in 11-phase LFH setting, PESCR improves accuracy by 4.5%, 2.8%, 2.3%, and 3.2% on Caltech-256, Food-101, Places-100, and ImageNet-100, respectively.

**Ablation study.** We investigate the effect of partial compression (by setting $\alpha > 0$) and diffusion-based data augmentation (by setting $p > 0$) on PESCR in Table 2. These two operations respectively focus on improving quantity and diversity of exemplars. For straightforward representation, we directly show the number of real ($R$) and synthetic ($S$) exemplars per class, instead of $\alpha$. The compressed ratio can be expressed as $\alpha = 1 - \frac{R}{b}$. It can be observed that when augmentation is not applied, the improvement by increasing exemplar quantity is relatively limited. When augmentation is applied, as the domain gap is reduced, the model benefits much more from additional exemplars. Jointly applying these two techniques yields the best result. However, excessive exemplars (when $\alpha > 0.3$) harm the model performance, as the model is biased towards learning from the dominating synthetic data.

Table 5: Average 11-phase LFH accuracies (%) of DER on three datasets with $b$ memory units/class, with and without PESCR plugged in. The accuracy improvements by applying PESCR are listed in the last row.

| | Food-101 | | | Places-100 | | | ImageNet-100 | | |
|---|---|---|---|---|---|---|---|---|---|
| | $b$=5 | 10 | 20 | 5 | 10 | 20 | 5 | 10 | 20 |
| DER | 77.3 | 79.1 | 80.4 | 67.1 | 68.6 | 69.5 | 77.9 | 79.3 | 80.2 |
| DER+PESCR | 80.1 | 81.3 | 83.2 | 70.4 | 70.9 | 71.8 | 81.1 | 81.4 | 83.4 |
| Improvements | +2.8 | +2.2 | +2.8 | +3.3 | +2.3 | +2.3 | +3.2 | +2.1 | +3.2 |

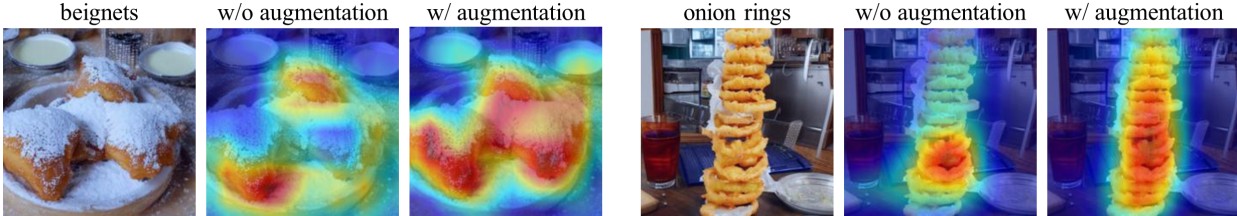

Figure 3: Two generated exemplars of Food-101 with their Grad-CAM given by the classification model trained without diffusion-based augmentation and with augmentation. Activation regions given by the model trained with augmentation capture the class objects much more accurately.

**PESCR with different CIL methods.** We incorporate PESCR with six different CIL methods and measure the accuracy increase in Table 3. PESCR improves the performance of five methods considerably by a large margin (from 3.2% to 16.1%), and has relatively weak performance only on FOSTER. As reported in (Wang et al., 2022a), increasing the number of exemplars per class from 20 to 50 in FOSTER brings almost no improvement, but with more than 50 exemplars, the domain gap leads to performance degradation for PESCR, thus the accuracy increase is not as tremendous as in other methods.

**Number of generated copies per image.** We explore the influence of increasing $K$ (the number of generated copies per image) on PESCR. Table 4 illustrates the accuracy improvement by generating more copies, thanks to the growing sample diversity. The performance increment gradually diminishes when $K \geq 5$, due to 1) the duplicated features generated from each edge map, 2) limited training epochs to learn from additional generations, and 3) the widened domain gap between real and superfluous synthetic images. Therefore, we use $K = 5$ copies per image for all the experiments.

**PESCR with different memory budgets.** To further understand the effectiveness of our approach under more limited memory limitations, we alter the memory budget $b$ and quantify the improvements of PESCR in Table 5. It is remarkable that our approach can consistently enhance CIL performance even with more restricted budgets.

**Grad-CAM visualization of generated exemplars.** To verify if the class objects generated by Control-Net can be successfully identified by the CIL model, we use Grad-CAM (Selvaraju et al., 2017) to detect the image region with the most importance to the classification decision. Displayed in Figure 3 are two synthetic exemplars of Food-101 with their activation maps produced by a model trained without diffusion-based data augmentation and a model trained with augmentation. Evidently, the augmentation process is vital for the classification model to comprehensively detect the generated objects. This ensures that the model can properly benefit from the synthetic exemplars during training in subsequent stages.

**Sampling scheme of prompt-based exemplars.** We study the effect of sampling order for choosing exemplars to store as prompts. As specified in Lines 30-31 of Algorithm 1, the default sampling approach is to pick the most representative $(1 - \alpha)b$ samples of each class to store as RGB images, while picking the next $24\alpha b$ representative ones to store as edges. We refer to this scheme as *least representative prompts*. Its alternate sampling scheme, namely *most representative prompts*, is to save the most representative $24\alpha b$ samples as prompts and preserve the next $(1 - \alpha)b$ ones. Another scheme is *random sampling*, which picks

Table 6:   Average and last 6-phase LFH accuracies (last accuracies in parentheses, %) of ImageNet-100 with different exemplar sampling modes. Augmentation probability $p = 0.4$, budget $b = 20$ units/class, and $R = 16$ real and $S = 96$ synthetic exemplars/class are saved. Each sampling mode is run three times and their average results are reported. (LRP: least representative prompts; MRP: most representative prompts.)

| Sampling Mode | LRP | MRP | Random |
|---|---|---|---|
| Accuracy | **83.7** (79.7) | 83.6 (79.6) | 83.6 (79.7) |

Table 7:   Total training time of DER in 2×A5000 GPU hours and average LFH accuracies (%) on 11-phase Food-101, with budget $b = 20$ units/class and augmentation probability $p = 0.4$. $R$ real and $S$ synthetic exemplars/class are saved in the buffer. The compressed ratio $\alpha = 1 - \frac{R}{b}$.

| #Exemplars/Class=$R + S$ | 20=20+0 | 43=19+24 | 66=18+48 | 89=17+72 |
|---|---|---|---|---|
| Compressed Ratio $\alpha$ | 0.00 | 0.05 | 0.10 | 0.15 |
| Training time | 22.3 | 25.9 | 29.5 | 34.1 |
| Accuracy | 81.9 | 82.0 | **83.2** | 82.4 |

$24\alpha b + (1 - \alpha)b$ most representative samples first, and then randomly chooses $(1 - \alpha)b$ to save as original images and $24\alpha b$ to save as prompts. We compare these three sampling schemes in Table 6. The three sampling modes yield similar accuracies, meaning that picking more or less representative images to be prompts does not have a significant impact on the final performance.

**Training time.** We provide the time to train a CIL model (with DER) in Table 7 with different numbers of exemplars per class. By compressing $\alpha = 10\%$ of the data, we can gain 66 exemplars per class (3.3 times the original). This costs approximately 32% more training time, but the model accuracy is substantially increased from 81.9% to 83.2%.

**Generation time.** Image generation based on diffusion model is relatively time-consuming, which is a limitation of our approach. For example, generating $K = 1$ synthetic copy of ImageNet-100 using 8 NVIDIA RTX A6000 GPUs takes approximately 15.5 hours. In this paper, we assume time is not a limiting factor in incremental learning, and we fully exploit the effectiveness of PESCR by increasing $K$. In practical applications, if a time constraint is imposed, $K$ might be reduced for more efficient generation.

## 5   Conclusion

In this paper, we propose PESCR, an exemplar super-compression and regeneration approach to enhance replay-based class-incremental learning methods by significantly increasing the quantity and diversity of exemplars under the same memory restriction. We challenge the conventional viewpoint that data from former classes can only be stored as RGB images, and present a novel prompt-based data storage approach: at the end of each incremental phase, the selected images are compressed to Canny edge maps, reducing memory consumption by a factor of 24. In subsequent stages, images of great diversity are regenerated from the edge maps and class tags by ControlNet, a pre-trained text-to-image diffusion model. To bridge the domain gap between real and generated exemplars, partial compression and diffusion-based data augmentation are introduced to let the model properly benefit from the synthetic exemplars. This enables us to directly apply ControlNet on our target datasets without any fine-tuning, so the generator does not have to be saved in our memory buffer. Comprehensive experiments demonstrate that our approach constantly improves the CIL model performance by a large margin on numerous datasets and CIL settings.

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

## A    Dataset Information

We present the detailed information of the datasets used in our experiments in Table 8.

Table 8:   Detailed information of the four datasets, including number of classes, total number of training/test images, average number of training/test images per class, and median image size.

| Dataset | Caltech-256 | Food-101 | Places-100 | ImageNet-100 |
|---|---|---|---|---|
| # classes | 256 | 101 | 100 | 100 |
| Total/average # training images | 21,436/84 | 75,750/750 | 300,000/3,000 | 128,856/1,289 |
| Total/average # test images | 5,472/21 | 25,250/250 | 10,000/100 | 5,000/50 |
| Median image size ($h \times w$) | $289 \times 300$ | $512 \times 512$ | $512 \times 683$ | $375 \times 500$ |

## B    Data Preprocessing

The image transformation procedures are listed in Table 9. Following FOSTER [45], we apply AutoAugment [9] in all CIL methods for a fair comparison. The same training transformations are applied to both real and generated images.

Table 9:   Training and test image transformations. Normalization has mean $[0.485, 0.456, 0.406]$ and standard deviation $[0.229, 0.224, 0.225]$.

| Training transformations | Test transformations |
|---|---|
| RandomResizedCrop(224), | Resize(256), CenterCrop(224), |
| RandomHorizontalFlip($p = 0.5$), | |
| ColorJitter(brightness=63/255), | |
| ImageNetPolicy(), | |
| ToTensor(), | ToTensor(), |
| Normalize(), | Normalize(), |

## C    Image Resizing

There are two approaches to transform an image ($h \times w$) into an edge map ($H \times W$), which can be in a different size to accommodate the input requirement of ControlNet. *Resizing edge map*: convert the image ($h \times w$) to an edge map ($h \times w$) and then resize the edge map ($H \times W$) by nearest neighbor interpolation. *Resizing image*: resize the image ($h \times w$) into an intermediate image ($H \times W$) by Lanczos interpolation and then convert it to an edge map ($H \times W$). We compare these two approaches qualitatively in Figure 4 and find that resizing the image can produce generations of higher quality. These two approaches consume similar memory in total.

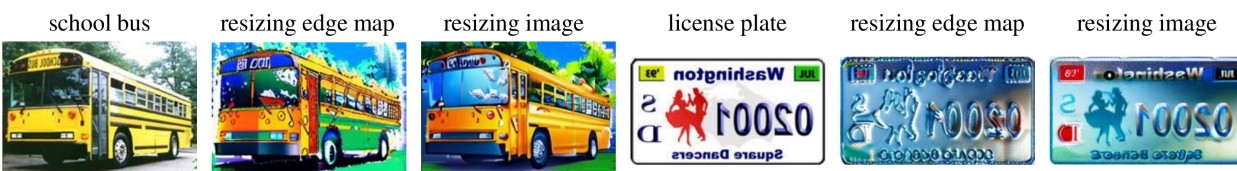

Figure 4:   Two example images from Caltech-256 and their generated versions by resizing edge map and resizing image.

# D    Image Generations

From each dataset, we show two images with their class labels, Canny edge maps, and generations by ControlNet in Figure 5. In general, the quality of generations is satisfying. ControlNet is able to generate diverse images simply by changing the random seed.

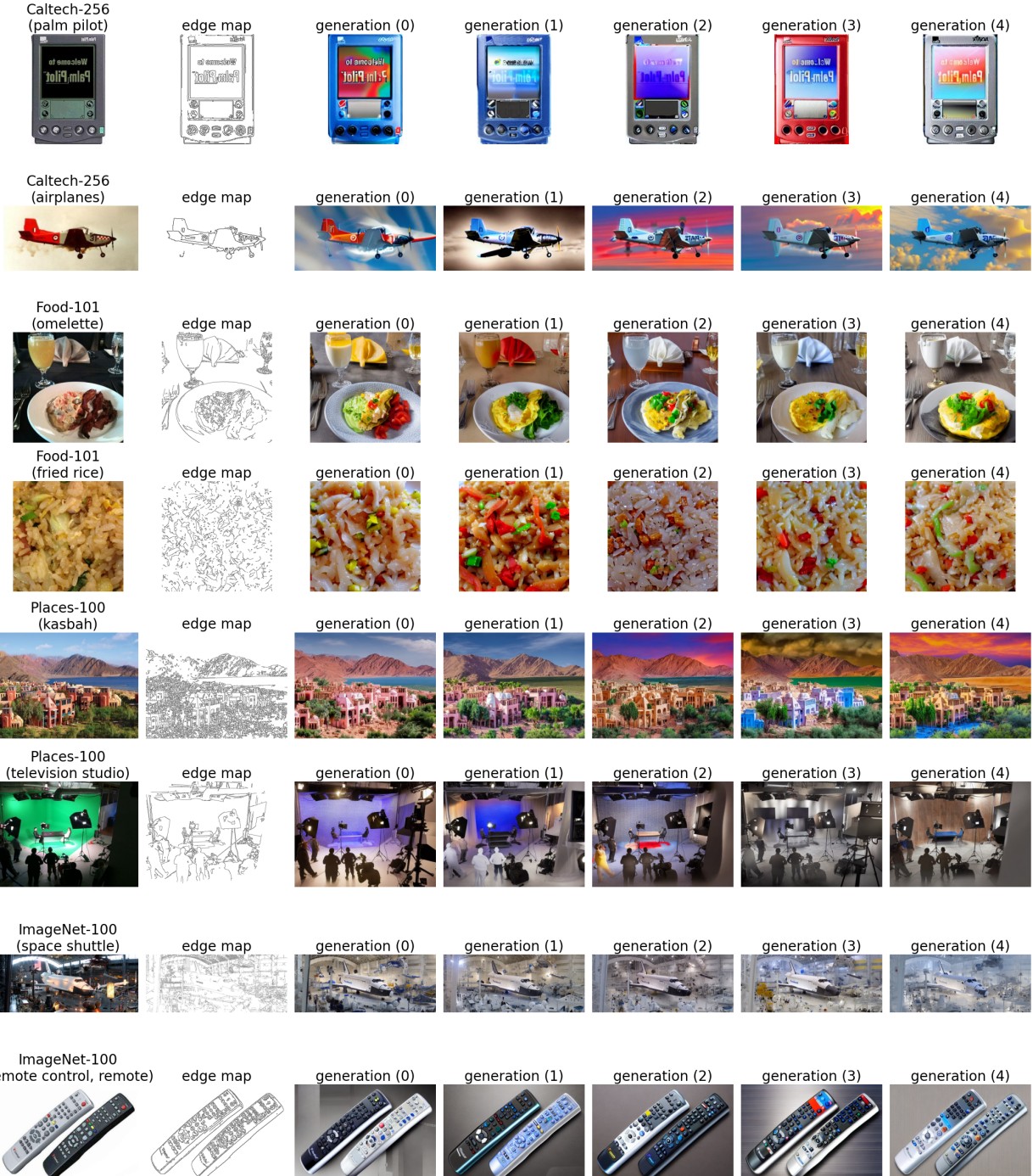

Figure 5:    Random images selected from the four datasets, with their class labels, edge maps, and five generations from ControlNet. Random seeds $0, 1, 2, 3,$ and $4$ are used to generate the five images respectively.

