# OpenReview forum: "Prompt-Based Exemplar Super-Compression and Regeneration for Class-Incremental Learning"
_TMLR — Rejected by TMLR_

### Review · Reviewer_pkpY · 2024-08-26

**Summary Of Contributions:**

This paper introduces ESCORT, a novel approach to class-incremental learning that addresses the limitations of exemplar storage in replay-based methods. By compressing images into visual and textual prompts, ESCORT significantly increases the number of exemplars that can be stored within a given memory budget. The method then uses a pre-trained diffusion model to regenerate diverse, high-resolution exemplars from these prompts during subsequent learning phases. Experiments across multiple datasets demonstrate that ESCORT consistently outperforms SOTA methods.

**Audience:**

Yes

**Broader Impact Concerns:**

ESCORT compresses images into visual and textual prompts, which are then used to regenerate exemplars. This process of compression and regeneration could potentially lead to unintended information leakage or privacy breaches if not properly safeguarded. A discussion on how the method ensures the privacy of the original data subjects is necessary.

**Claims And Evidence:**

Yes

**Requested Changes:**

- A detailed analysis of the computational overhead introduced by using a pre-trained diffusion model for exemplar regeneration, which is crucial for understanding the practical applicability

- Expand comparisons with non-replay-based CIL methods to provide a broader context for ESCORT's contributions.

-  Consider including other relevant metrics such as forgetting rate, learning speed, or computational efficiency alongside accuracy.

**Strengths And Weaknesses:**

Strengths:
- It introduces an innovative method for exemplar compression and regeneration in class-incremental learning

- The compression technique reduces memory consumption, allowing for more exemplars to be stored within the same memory budget.

- Experiments across multiple datasets demonstrate ESCORT's effectiveness

Weaknesses:

- The paper does not thoroughly discuss the computational overhead of using a pre-trained diffusion model for exemplar regeneration, which may be significant?

- It might benefit from more extensive comparisons with non-replay-based CIL methods to provide a broader context for its contributions.

- While accuracy is used as the primary metric, it could benefit from exploring other relevant metrics such as forgetting rate, learning speed, or computational efficiency.

---

> ### Author Response · Authors · 2024-09-20
> **Response to Reviewer pkpY**
>
> 1) Computational overhead of exemplar regeneration.
> An extra paragraph (“generation time”) has been added to Section 4.2 to discuss the time needed for exemplar generation. Image generation is based on ControlNet, and the computational overhead is the same as ControlNet. Although generation is relatively time-consuming, we assume, in each incremental phase, users are given sufficient time for exemplar generation, and we focus our study on maximizing model performance. In practice, users can spend less time on generation by controlling the number of generations per image (the parameter K in the paper, with its effect on performance shown in Table 4), if time is a concern in the application.
>
> 2) Non-replay-based CIL methods.
> Our approach enhances replay-based CIL methods by compressing exemplars, thus for fairness, comparison is made mainly with replay-based methods to keep problem setting (memory budget) consistent. This follows the previous convention of evaluating replay-based approaches (e.g., DER, FOSTER, CIM, MEMO). Non-replay-based approaches, without having access to former class data, are inherently disadvantageous when compared with their replay-based counterparts.
>
> 3) Other metrics.
> Computational efficiency alongside accuracy has been discussed in Section 4.2 and Table 7. As for the model performance metrics, we would like to focus on the most representative metric (accuracy) for evaluation, as adopted in most CIL papers.
>
> 4) Broader impact concerns.
> The privacy issue is a general concern for all replay-based CIL methods, not specifically for our method. Our approach, by saving RGB images as edge maps, only mitigates privacy problems as private information is greatly obscured in edge maps.

---

### Review · Reviewer_bqbK · 2024-09-10

**Summary Of Contributions:**

This paper proposes a Class Incremental Learning (CIL) approach, called ESCORT. In the CIL setting studied in the paper, 1) memory for storing the dataset is fixed 2) data for new classes is streamed. The algorithm presented proposes to maximize the utility of the fixed buffer by using generative models over image datasets. The base algorithm converts (image, label) pairs into (edge map, prompt) pairs, yielding a 24x compression rate for the images. The algorithm is further extended to 1) retain some original samples in RGB space 2) apply diffusion based data augmentation to compressed pairs to increase generalization. ControlNet over Canny edge maps are extensively used in the implementation. Results are evaluated on Caltech-256, Food-101, Places-100, and ImageNet-100 and compared to 7 CIL methods.

**Audience:**

Yes

**Claims And Evidence:**

Yes

**Requested Changes:**

See weaknesses above.

**Strengths And Weaknesses:**

**Strengths**:
* The algorithm seems simple and effective. It is easy to understand when reading.
* Improvements are reported over multiple CIL methods.
* Ablations give some sense of where performance improvements and costs come from.

**Weaknesses/Suggestions**:
* [Critical] "Due to data privacy and hardware limitations in real-world applications, the memory to store exemplars in CIL problems is often highly restricted, sometimes far less than 20 units per class." -> Can you provide a citation?
* [Recommended] The ablations seem to be limited in scope (e.g., fixing a hyperparameter and dataset as in Table 4, or only using a subset of the 4 datasets in Tables 3, 5, 6, and 7). Figure 3 is also only 2 examples. A more complete sweep of the possible grid would be appreciated.
* [Recommended] ImageNet-100 is used rather than ImageNet-1000, which would be more convincing. It would also strengthen the work to compare to datasets with a domain shift (i.e., not photo-realistic scenes).
* [Recommended] Depending on the audience and context, the acronym of the method may be viewed as inappropriate. Replacing it with a difference acronym would avoid confusion.
* [Critical] "Herding" is mentioned and cited but not described further. Can you add a short inline explanation?
* [Recommended] Nit: End quotes in "Textual prompt extraction." are not properly formatted.
* [Critical] Can you comment on lossless coding e.g., entropy coding? For example, can edge maps be losslessly compressed?
* [Critical] Can you clarify what the logical disjunction operator is doing (e.g., is it a set intersection of the sets used in Equation 4)?
* [Critical] Algorithm 1 would benefit from further structure, such as color coding and annotations.
* [Critical] Can you provide the inference latency of a regenerated example and how it compares to the training time latency of the underlying ResNet-18 model?

---

> ### Author Response · Authors · 2024-09-20
> **Response to Reviewer bqbK**
>
> 1) Citation.
> We have changed the wording of that sentence to make it less absolute/specific. The main point of that section is to discuss the performance of our approach under more limited memory budgets.
>
> 2) Limited ablations.
> Reporting results on all datasets under all settings is regarded as unnecessary, as we hope to keep the paper concise and succinct.
>
> 3) Datasets.
> We already tested our approach on four different datasets, covering a variety of scenes. We believe the results have illustrated the effectiveness of our method on different settings.
>
> 4) Method naming.
> The acronym is now changed to PESCR to avoid confusion.
>
> 5) Herding.
> An explanation of herding is now added to Section 3.4. Readers are welcome to go to the reference for details.
>
> 6) End quote format.
> The end quotes are now properly formatted.
>
> 7) Lossless coding.
> Entropy coding techniques such as Huffman coding and arithmetic coding can indeed further compress edge maps, but they can also be applied to RGB images, which are essentially bits as well. The effect of lossless coding on edges/images depends on the coding algorithm and various image properties, which are not the focus of this paper.
>
> 8) Logical OR operator.
> The operator means in each training iteration, only one sample from the two sets (either one but not both) is selected for model update.
>
> 9) Algorithm 1.
> The algorithm is improved.
>
> 10) Inference latency of regenerated examples.
> Classification inference speed is the same for regenerated and original images, since the image size is not changed. The time for regeneration based on ControlNet is now added to Section 4.2.

---

### Review · Reviewer_niCB · 2024-09-11

**Summary Of Contributions:**

This manuscript introduces a generative replay-based algorithm, ESCORT, for Class-Incremental Learning (CIL), designed to address the limitations of direct replay methods, particularly regarding memory constraints and exemplar diversity. These limitations in CIL result in storing a limited number of exemplars and lead to imbalanced (or even biased) learning between old and new classes. To mitigate this, ESCORT compresses RGB exemplar images into Canny edge maps and text prompts, allowing up to 24 times more edge maps to be stored within the same memory budget compared to storing original images. The authors leverage an off-the-shelf diffusion model (e.g., ControlNet) from the public cloud to regenerate diverse, high-quality exemplars from the stored edge maps and text prompts. Additionally, the authors introduce partial compression and diffusion-based data augmentation to address the domain gap between real and synthetic exemplar images. This approach enables the storage of a larger number of exemplars, resulting in improved CIL performance across multiple benchmarks.

**Audience:**

Yes

**Broader Impact Concerns:**

I do not identify any significant ethical concerns or broader impact issues that would require the addition of a Broader Impact Statement.

**Claims And Evidence:**

Yes

**Requested Changes:**

Although this manuscript shows the effectiveness of the use of the generative models for CIL, I believe this is not new as the generative-replay CIL methods were introduced many times in the literature.

- **Novelty and Positioning in the Literature:** The paper would benefit from a more explicit discussion about how ESCORT differentiates itself from existing generative replay-based methods (such as VAE, GAN, and diffusion models) beyond the use of the {edge maps, ControlNet-Canny} pairing. Although the use of these components is different, it would strengthen the manuscript if the authors provided a deeper explanation of how this approach fundamentally advances the field, rather than being perceived as merely updating a version of the generative model of the previous generative-replay method. This would help address concerns about the novelty of the contribution
- **Comparison with Generative-Replay Baselines:** Including direct comparisons with generative replay baselines would be required. While the manuscript highlights ESCORT’s superiority over direct replay methods (which store original RGB images), it is equally important to demonstrate how ESCORT performs relative to, and distinct from, existing generative replay methods. A quantitative comparison in this regard would provide stronger evidence of ESCORT's efficacy.
- **Addressing Domain Gap and Computational Costs:** The current manuscript points out the potential domain gap between real and synthetic exemplars but does not provide sufficient experimental evidence regarding the efficiency of the partial compression and diffusion data augmentation strategies in mitigating this gap. More empirical results demonstrating the effectiveness of those algorithms (compared to other generative-replay methods) would strengthen this section.

**Strengths And Weaknesses:**

**Strengths:**
- **Memory efficiency:** ESCORT significantly enhances memory efficiency, allowing for the storage of more exemplars within the same memory constraints.
- **Increased diversity:** By leveraging a pre-trained diffusion model, ESCORT effectively generates diverse exemplars, addressing the overfitting issue in old classes due to limited exemplar diversity.
- **Broad applicability:** The approach demonstrates consistent improvements across various datasets and CIL setups, showcasing its versatility.
- **Broad compatibility:** ESCORT integrates smoothly with existing CIL methods, offering substantial performance improvements across different (direct) replay-based algorithms.

\
**Weaknesses:**
- **Novelty:** The key question raised by the authors - ``Is it efficient to save old-class information as RGB images?`` is not new, as many generative replay-based methods already exist that utilize generative models instead of directly saving the original RGB images. For instance, not only VAE [1] and GAN [2-9], but also diffusion models [10-12] have been used in the literature. Although the use of pairs {edge maps, ControlNet-Canny} is new, I am concerned that this manuscript revisits the existing literature by updating the generative model with newer methods.
- **Comparison with Generative-replay baselines:** Although the authors mentioned generative replay methods in the related work section, there is no comparison with any generative-replay baselines. As allowing more exemplars obviously can enhance direct-replay baselines (saving the original RGB images), I suspect generative-replay methods usually can improve the CIL performances compared to direct-replay baselines. Therefore, only showing applying a generative CIL method to previous direct-replay baselines is not enough as it would be already validated across the literature. However, I can not find any comparison or discussion that the proposed method is superior to the previous generative-replay CIL methods.
- **Domain Gap Reduction:** First, as pointed out by the authors, the effectiveness of the sorting algorithms for partial compression is not significant. Second, the authors do not consider and discuss the cost of diffusion model usage. For instance, not only the cost of K generations from each edge map but also diffusion-based data augmentation requires K number of generated copies for all real images. It would require pre-processing before every training phase to learn $\Theta_i$, which increases each training cost (for generation and holding samples) - however, Table 7 does not mention it, except for training time after this pre-processing.
- **Typo:** What is $\alpha$ in eq (4)? Is it $\gamma$ in eq (3)?
- **Question:** The authors assume that the same pre-trained diffusion model can be directly downloaded from the public cloud at any time when necessary, so there is no need to store the fine-tuned generator in memory within the CIL framework. However, if we have access to the public cloud, why not access the data (exemplars) potentially uploaded to the public cloud as well? It would be helpful if the authors could provide further thoughts on the setup of public cloud access within the CIL framework.

[1] Jiang et al., Ib-drr-incremental learning with information-back discrete representation replay. In CVPRW, 2021.

[2] Shin et al., Continual learning with deep generative replay. In NIPS, 2017.

[3] He et al., Exemplar supported generative reproduction for class incremental learning. In BMVC, 2018.

[4] Hu et al., Overcoming catastrophic forgetting for continual learning via model adaptation. In ICLR, 2019.

[5] Kemker et al., Fearnet: Brain-inspired model for incremental learning. In ICLR, 2018.

[6] Ostapenko et al., Learning to remember: A synaptic plasticity driven framework for continual learning. In CVPR, 2019.

[7] Xiang et al., Incremental learning using conditional adversarial networks. In ICCV, 2019.

[8] Wang et al., Ordisco: Effective and efficient usage of incremental unlabeled data for semi-supervised continual learning. In CVPR, 2021.

[9] Cong et al., GAN Memory with No Forgetting, In NeurIPS, 2020.

[10] Gau et al., Ddgr: continual learning with deep diffusionbased generative replay. In ICML, 2023.

[11] Jodelet et al., Classincremental learning using diffusion model for distillation and replay. In ICCVW, 2023.

[12] Kim et al., SDDGR: Stable Diffusion-based Deep Generative Replay for Class Incremental Object Detection, In CVPR, 2024.

---

> ### Author Response · Authors · 2024-09-20
> **Response to Reviewer niCB**
>
> 1) Novelty.
> The previous generative-replay methods, including the ones you mentioned, have at least one of the following issues, while our approach tackles these problems all at once.
> Regenerated images are blurrier or in a lower resolution than the original ones (lossy compression).
> One or multiple generators are fine-tuned on the dataset, making it necessary to store the generator(s) using the memory. Many of those works just take it for granted and calculate memory consumption only by the number of images stored, without taking the stored generator(s) into account at all.
> Employ a pre-trained diffusion model (though publicly available) to directly generate images without introducing the compression technique to store more data within the same memory budget, inefficiently utilizing the memory buffer.
> In addition, TMLR states that novelty of the method is not a necessary criteria for acceptance (https://jmlr.org/tmlr/acceptance-criteria.html).
>
> 2) Comparison with generative-replay methods.
> There are generally two types of generative-replay methods.
> One requires training the generator using their datasets. They do not count the generator into their memory, leading to an unfair comparison with replay-based methods without any generator. Our approach follows the memory consumption of direct-replay methods, without the need to store any fine-tuned generator using our buffer.
> One uses pre-trained models for image generation, producing samples without any relation to the training images. Our generated images are based on edge maps of the original dataset, producing more relevant images.
>
> 3) Domain gap reduction.
> 3.1) Sorting is indeed not significant. We firstly propose a plausible strategy in Section 3.4 to select which images to store as edge maps. We assume it would be reasonable to store less representative ones as edge maps, so that the representative information will remain intact in the buffer, which intuitively makes sense. But eventually, according to experiment results, the selection strategy does not matter too much.
> 3.2) The time for image generation based on ControlNet is now being discussed in Section 4.2.
>
> 4) Typo.
> The typo in Equation (4) is now fixed. Thanks for pointing it out.
>
> 5) Memory storage assumption.
> In this paper, we assume two types of memory storage.
> Public cloud: remote public storage, already deployed/established, always available, free usage, no privacy. Usually used to store general-purpose models pre-trained on large-scale public datasets. E.g., pre-trained ControlNet. Due to privacy issues, the exemplars of our dataset should not be stored in the public cloud.
> Private storage: used to store our own data, which might contain private information. It can be either local or remote, usually with a limited storage buffer.

---

### Decision · Action_Editor_HDxL · 2024-11-15

**Recommendation:** Reject

**Comment:**

This paper proposes a method for CIL based on a generative replay algorithm to address the limitations of exemplar storage and memory constraints in direct replay methods. The main idea is to leverage a pre-trained diffusion model for CIL by generating exemplars. First, the authors utilize ControlNet with saved Canny edge maps from real exemplars to generate high-quality images for training. Second, the authors propose diffusion-based data augmentation to mitigate the real and synthetic gap by replacing real training samples with more generated copies. Two of the three reviewers remained negative after the rebuttal; however, all reviewers raised concerns about insufficient empirical evidence (as the authors did not provide additional supporting results) and computational overhead from diffusion-based generation. AE shares these concerns, agreeing that the current state is not well-supported by sufficient empirical evidence and that concerns surrounding computational feasibility are not fully addressed. Addressing these issues would require a substantial amount of additional work, and AE cannot recommend acceptance in the current state.

**Audience:**

Yes

**Claims And Evidence:**

No, all reviewers pointed out that empirical evidence is not sufficient, and AE also agrees with this.

**Resubmission Of Major Revision:**

The authors may consider submitting a major revision at a later time.